# The Role of Landscapes and Landmarks in Bee Navigation: A Review

**DOI:** 10.3390/insects10100342

**Published:** 2019-10-12

**Authors:** Bahram Kheradmand, James C. Nieh

**Affiliations:** Section of Ecology, Behavior, and Evolution, Division of Biological Sciences, UC San Diego, La Jolla, CA 92093, USA; jnieh@ucsd.edu

**Keywords:** landmark detection, salience, visual information, multimodal navigation, *Apis mellifera*

## Abstract

The ability of animals to explore landmarks in their environment is essential to their fitness. Landmarks are widely recognized to play a key role in navigation by providing information in multiple sensory modalities. However, what is a landmark? We propose that animals use a hierarchy of information based upon its utility and salience when an animal is in a given motivational state. Focusing on honeybees, we suggest that foragers choose landmarks based upon their relative uniqueness, conspicuousness, stability, and context. We also propose that it is useful to distinguish between landmarks that provide sensory input that changes (“near”) or does not change (“far”) as the receiver uses these landmarks to navigate. However, we recognize that this distinction occurs on a continuum and is not a clear-cut dichotomy. We review the rich literature on landmarks, focusing on recent studies that have illuminated our understanding of the kinds of information that bees use, how they use it, potential mechanisms, and future research directions.

## 1. Introduction

### 1.1. What Are Landmarks?

A major and fascinating question in the study of animal cognition is how animals identify, remember, and use landmarks to navigate. Reliable navigation is crucial for animals that migrate or have nests, hence, and multiple species have evolved different strategies to find their way to food sources and back to their home. Sophisticated strategies and neural mechanisms have therefore evolved to facilitate navigation. For example, depending on their natural history and ecology, many species need spatial memories to guide them on their paths. These navigational strategies use multiple sensory modalities, including vision, olfaction, proprioception, electroreception, and magnetoreception. A recurring theme is that animals usually orient themselves towards their goals with multiple, redundant forms of information and prioritize this information according to its reliability and salience (relative conspicuousness to the observer in a given motivational state) [1].

For example, each environment offers multiple pieces of visual information, ranging from the position of celestial objects above the skyline to the colorful petals of a flower. Animals can use these features to orient themselves in the correct directions at different points along their routes. As an animal explores its surroundings, it comes across entities that are salient in some manner. The presence of these entities is used to help identify and define the location at which they are found [2,3]. It is not known how animals perceive their goals relative to or nested within an overall landscape, but animals functionally use these entities at different stages along a path to reach a goal. These entities are classically referred to as landmarks [4]. In contrast, we reserve the term landscape to refer to more global cues that are the combination of multiple landmarks. The dichotomy between landmarks and landscapes is not absolute and depending on the exact sensory stimuli, animals may perceive them as a continuum.

In many cases, it is unclear if a single stimulus aspect is enough to describe an entity or if entities consist of multiple stimuli that animals can recognize and use. The latter is quite likely since entities are real world objects with multiple characteristics. In the study of animal navigation and, more recently, machine vision, the term landmark has been used in multiple contexts with different meanings for different organisms [5]. Here, our goal is to review the different senses in which the term landmark has been used and how landmarks and landscapes are used by honey bees. However, we find it useful to include some studies of other insects that navigate using landmarks in order to compare and contrast with honey bees [6,7]. We focus primarily on examples drawn from vision but have tried to define our overall concepts broadly to encompass multiple sensory modalities.

### 1.2. How Have Landmarks Been Described? 

One of the earliest questions posed by researchers was if a location is recalled in terms of the overall visual input at that location, or if a small number of distinct visual stimuli are enough to identify a location [3]. In other words, animals may recall specific places based upon a visual gestalt (an organized whole that is greater than the simple sum of its parts: the gestalt hypothesis), or can they use a smaller subset of the visual information available (key feature hypothesis). Early experiments by Tinbergen showed that digger wasps use local landmarks close to the nest entrance while disregarding more distant landmarks [8]. In this case, and in multiple others, the entire visual gestalt is not necessary. Similarly, the classical way in which honey bee researchers [9] train foragers to a food source by moving a feeder over multiple forager visits to the desired location exploits the preference of foragers for familiar local landmarks (the visual appearance and odor of a feeder and its surrounds) over more distant landmarks. 

Relevantly, Anderson [10] showed that a rather simple model consisting of only a few landmarks in different parts of a bee’s field of view can explain the searching behavior of the bee in an experimental arena, reinforcing the claim that simplified features, not an entire complex landscape, are sufficient for successful navigation. Bees can, therefore, simplify a complex visual scene by extracting key features such as color, size, orientation, and distance. However, multiple elements, including the landscape, can also play a role. Cartwright and Collett [11] showed that bees returning to a food source tend to search for it in areas where the overall image projected on their eyes most closely matches their memories, a process referred to as “picture matching”. They increased the size of the landmarks and showed that bees would search at a farther distance where the apparent size of the landmarks was the same as they experienced during their training. 

As noted above, landmarks comprise a very diverse category of objects. Some goals such as the hive and food sources may be conspicuous enough to be regarded as landmarks themselves. However, some goals may be relatively inconspicuous until the animal approaches them and thereby require guiding landmarks along the way. A potentially useful classification for landmarks is, therefore, to distinguish them by how their size changes as the observer moves. Some landmarks will significantly change in apparent size (as defined by the visual acuity and discrimination ability of a given animal) as the animal approaches (closer landmarks) and others such as mountains or certain features at the horizon will not appreciably change in apparent size over the route (distant landmarks). These very distant landmarks are part of the overall landscape [12,13] and have also (under a slightly different context) been called “global cues” [14]. 

In our definition, we, therefore, classify celestial objects as part of the landscape, not as a landmark per se. We recognize that there is disagreement about how to classify celestial objects that guide navigation. Dyer and Gould [15] call the sun a celestial landmark. Since the sun moves, this definition is at odds with Horridge [2] who argues that the landmark gives identity to a specific location. Rather than create new terms, we instead define “near” versus “far” landmarks using the distinction of perceived size changes. We note that this concept is not limited to vision, but could apply to other senses that experience substantial change (“near”) or little change (“far”) over the scale of an animal’s route, namely magnetoreception [16] and chemosensation [17]. 

This idea of classifying landmarks by using changing stimuli relative to the receiver is not new. Collett and Harland [18] defined two types of landmarks, both of which provide stimuli that change as the receiver approaches: “isolated” landmarks are singular entities that can be reached and rapidly passed. In contrast, “boundary” landmarks are continuous and the animal can move alongside or through them [19]. This approach suggests that panoramic landscape information helps with guiding an animal’s overall orientation, whereas reachable landmarks define points in space. Bees distinguish between very similar views of a complex natural landscape [20] or similar artificial stimuli [21]. It is not clear if bees generalize different views of the same landmark based on a cue that is recognizable from any direction. The alternative is that a landmark could only be used in navigation when perceived from a specific angle. 

The relationship of a landmark to the goal is another important aspect of defining landmarks. This relationship could be in terms of distance, a specific orientation, or a specific order of appearance between the goal and one or more landmarks [22,23]. If the goal is much less conspicuous than its surrounding objects, the relationship between the landmark and the goal is a key factor in defining the location, whereas if the goal is comparably conspicuous, it may be the only visual cue needed to find it within a short range. 

### 1.3. Why Is It So Difficult to Define Landmarks?

It is difficult to pinpoint what animals learn in a visual learning assay when multiple aspects of a stimulus may be salient. Bees can use a single object (or a specific arrangement of a few objects) as a landmark, but objects have multiple features, and the feature that a bee uses can depend on other objects in the environment. This occurs because a cue is useful when it helps to distinguish the correct behavioral choice from an incorrect one, and this depends on the informational content of the scene [24]. Hertz [25] showed that bees trained to a feeder with a specific pattern would still land on and feed at a feeder with a different pattern if it shared some common features with the original pattern. Dyer et al. [20] demonstrated that such acceptance happens with complex images, but if two images were offered during training, one always paired with reward and the other with lack of reward, bees learned to distinguish highly similar images. 

### 1.4. A Proposed Definition for Landmarks

We define a landmark as an object or set of objects with useful and salient features that animals in a given motivational state can use to navigate towards a goal. What features are useful depends on the landmark and its immediate surroundings, since the utility of these features (individually or combined) is a function of their uniqueness, conspicuousness, stability [26], and context. Context includes both the relationship of objects to each other within a landscape (inter-landmark context) and the relationship of features’ information, potentially covering multiple sensory modalities, for a given object (intra-landmark context). For example, solar information is generally useful and reliable for navigating bees, and honey bees prioritize using the position of the sun, deduced from direct sighting or via sky polarization patterns (henceforth referred to as e-vector) to orient. However, landmarks also help bees orient, particularly when other sources of information are unavailable [12,27]. 

Uniqueness and conspicuousness of features allow easier recognition of the landmark. Unstable, moving objects do not typically serve as landmarks unless they are the goal (such as a moving feeder during training, an admittedly non-natural context). General landmarks that are present throughout a landscape (a river or a hedge) may be used differently from local (point) landmarks at the goal [4]. Menzel et al. [19] show multiple examples of guidance by continuous landmarks, in line with older experiments [27,28] in which bees prioritize using landmark information over celestial cues (sun and *e*-vector) when the hive is located near such landmarks. A diverse set of experiments show that the effect of an elongated landmark such as a forest edge is strongest when compass direction and the visual features of the landmark are congruent. This effect supports Collett and Harland’s [18] suggestion that landmarks can be classified as isolated/local vs. boundary/context. It is unclear whether landmark identity is defined solely based on visual (and other sensory) features or, in part, by the relation to specific locations in space. 

## 2. Representations of Space in Honeybees

A location in space can be defined in multiple ways. The most abstract way, used by humans and perhaps some animals who perform long-range migrations, is to define absolute space. This requires using reference points that are large-scale stimuli such as the earth’s magnetic field or its rotational axis and thus require very accurate sensory input to measure the positions of the global reference points. At a smaller scale, space can be represented with reference to smaller stimuli, such as hills or trees. When an animal needs to learn more than one location, it can either learn visual properties of each location separately and, potentially, learn a direct route between them. This mechanism is referred to as specialized route memory or egocentric navigation, and the animal knows where it is based solely upon its current view. 

Another possibility is that the animal also forms a connection between the two locations, forming a vector with a specific direction (compared to global or local reference points) and a specific distance, so that it can navigate between those two points without having to learn the details of the route that connects them. This mechanism is known as generalized landscape memory [29], or allocentric navigation. Why might generalized landscape memory be beneficial? As more locations are learned, if they are learned separately, the animal can only travel between them using known paths. However, if they are learned in connection to each other, it is possible that knowing the spatial relationship between multiple locations allows the animal to make shortcuts and travel novel routes between known locations (interpolation) or to find the way to a known location from an unknown location (extrapolation). 

Tolman [30] proposed the concept of a cognitive map, a system of coding space which uses relational information to build a representation. This representation could “link together conceptually parts of an environment which have never been experienced at the same time” [3] and several species, including bees, are claimed to possess cognitive maps [31,32,33]. Some experiments provide evidence that bees take such novel shortcuts [34,35]. However, in some cases, animals may be using simpler mechanisms [33] (see below) to perform the same tasks, and it could be difficult to reject alternative hypotheses. When other strategies could account for novel shortcuts [36], cognitive maps are difficult to demonstrate given that multiple experiments are required to exclude all other possibilities [37,38], and this hypothesis remains actively investigated and debated. Research on humans suggests that Euclidean space is not naturally derived from visual experiences and that a cognitive graph made from paths between locations explains human navigation better than a cognitive map [39].

Honeybees are a unique model organism for studying spatial representations since their waggle dance depends on the visual memories that they form on their trips. Follower bees also develop a spatial representation based on the tactile input from the dancers [40]. Whether all three codings of space (guiding the forager’s navigation, the dancer’s movements, and the follower’s exploration) are rooted in the same spatial representation is yet to be determined. 

## 3. Different Types of Navigation

Honeybees navigate using disparate visual cues at different parts of their route. Upon exiting the hive, the bee has access to limited stimuli: landmarks around the hive and direction information from the *e*-vector. She can use these cues to travel in the correct direction. On her route, she can compare the real-time visual input of near and far-range landmarks to memories of points on the route on her previous trips and maintain her heading while keeping track of how far she has travelled. Once near her floral destination, she can use short-range landmarks to pinpoint her target [41]. These navigational strategies broadly fall into three categories: alignment imaging matching, positional image matching, and path integration [41]. At each part of the path, if the necessary visual information is available, bees can use it to navigate.

Alignment image matching provides a simple way of comparing current views or cues extracted from a view with memories from previous trips. The animal then tries to maximize the match through corrective movements. If the bee is not too far from known areas where familiar features are abundant, this strategy allows her to stay on course. Positional image matching works by extrapolating the current location from differences between relationships among parts of the current view image and the same relationships in memories of views. The only requirement for positional image matching is that the panoramas (landscapes) of the novel area and the known areas share sufficient items in common. Alignment and positional image matching are not necessarily achieved in different ways, rather, they are different scalings of using memorized visual information to orient towards a goal. Path integration does not require landmarks, but only a source of the directional reference point (offered by the sun and its *e*-vector) and a source of odometry (optic flow or stride counting). By keeping track of the path segments taken, the animal can ideally integrate its current location at any moment and find a homing vector [41]. 

The relative accuracy of these three strategies depends on the situation. Path integration has a relatively low error at shorter distances and in novel areas but can result in larger errors when orienting near the goal because the remaining vector is much smaller than the accumulated error. Alignment image matching requires familiarity with the route but can guide the animal with minimal error, and can be learned in a single trip. Landmarks clearly play key roles in alignment and positional image matching strategies, but can also be combined with path integration to create short vectors for segments of the entire route [9]. We will discuss the contributions of landmarks to each of these strategies below.

### 3.1. Snapshots

A snapshot is a set of extracted features of part of or the entire visual field (and perhaps other sensory inputs) at a location. Although it is possible that an image is memorized as a whole, it is more likely that only its prominent features are extracted and memorized. Bees learn sets of visual snapshots [42] and use them in different ways to guide their behavior at different points on their trips [43], like ants [44]. When using snapshots, a landmark can help a bee to recall a vector or to maintain a steady heading by holding the landmark at a position in its field of view (preferably frontal) by flying closely to the left or right of it. 

Collett and Harland [18] argue that a landmark does not necessarily lead to the recall of a certain location and that panoramic contexts are more reliable than single landmarks that are single entities because using panoramic contexts could reduce the risk of misidentifying or missing a landmark. What a context consists of is not explicitly defined, but could be quite broad, including the time of day and motivational state [45]. However, their experimental design suggests that the general patterning of the environment (stripes versus dots) can create a context. Perhaps large and distant objects could also serve as contexts.

### 3.2. Vectors

A vector is a linear path starting from a specific point and ending at a destination. It has a given distance and direction relative to the reference points that the animal uses, such as landmarks or the *e*-vector. For example, ants recall different vectors depending upon the visual landmarks that they encounter [46].

Vectors can be used in multiple strategies. A vector could connect the nest and different food sources or lead from each of them to mid-path points from which other vectors could provide guidance. An appropriate panoramic context could activate a local vector [18], and a sequence of these vectors would then allow the animal to travel a non-linear path by segmenting this path into shorter linear sub-paths. Alternatively, bees can learn complex flight maneuvers and perform them in a fixed sequence [47]. 

Using vectors does not exclude using snapshots because the two concepts are complementary. Moving in the correct direction will inevitably lead the animal to the next snapshot location where the next vector can be activated without a need for the last vector to be completely executed. Experiments with bees flying in a tunnel show that foragers looking for a missing feeder search in the correct distance after passing the last landmark on their route [48] and that the total image motion (optic flow) determines the distance communicated in the waggle dance [9,49]. Similar behavior is observed in desert ants when multiple path segments of different lengths and directions are used to reach a food source [50,51], although it is difficult to exclude the role of landmarks at the turning points. 

### 3.3. Features and the Sensory World of the Animal

Although we have defined landmarks (see above), a strict definition is elusive because there is no consensus on what the brains of different animals consider to be the signature or signatures of a location or object. Bees have a very capable visual system, one that has evolved to solve multiple problems. However, it is difficult to provide bees with stimuli to dissect perfectly their navigational strategies. The gestalt and snapshot hypotheses suggest that multiple parts of a scene can help code space. Specific cues within an image, which we refer to as features, could be extracted separately through multiple feature detectors, and the coincidence of these features may encode the memory of a location. Examples of features include total area, center point of an area, total edge length, average local orientation, radial or tangential edges, colors, etc. However, anatomical and physiological studies of the eyes and optic lobes of insects have elucidated some of the feature extraction mechanisms, and behavioral tests have dissected the relative importance of different features and how animals use them in hierarchies. 

Two hypotheses about what visual information bees remember are parameter-based memory and pictorial memory [52]. In parameter-based memory, a series of measurements of different features are summed together to create a final score for a scene. Later, if a scene has a similar score, it indicates a match. Pictorial memory, however, must encode the relative position of the elements of the scene. Gould [53] argued that since relative position plays a role in pattern recognition, pictorial memory, and not parameter-based memory, is how bees memorize scenes. As an intermediate hypothesis, it is possible that different areas of the eye sense the presence of one or a few specific cues in a parametric manner and that matching occurs when several areas simultaneously receive input sufficiently similar to the memorized information [54].

### 3.4. Feature Hierarchies

Many experiments have tried to tease out the relative importance of different visual features. Gould and Towne [55] reviewed the relevant literature during the 1960–1980s and provide a list of features that bees prefer to use in order of presumed importance: (1) relative color inputs of areas, (2) the distribution of line angles, and (3) and the distribution of borders.

Horridge [2,9] offers a different prioritized list based on several experiments with primarily black and white patterns: (1) total area, (2) position of the center of area, (3) total modulation, (4) radial edges, (5) average local orientation, (6) positions of hubs, and (7) tangential edges. Large black circles were preferred over radial spokes or parallel edges, and symmetry in a pattern of bars was preferred over the edge orientations of symmetrical patterns. Recent studies demonstrate the ability of honey bees to learn complex visual tasks using different categories of visual stimuli [56]. 

When landmark information and vector-based navigation provide conflicting information, landmarks seem to be more important [48,57,58,59]. Prominent landmarks can override *e*-vector information [28,60], and it is possible that landmark-based directional information (computed on the return flight) affects the navigation and the waggle dance of foragers [61]. Motivation could play a role in how foraging bees infer and use their spatial information. When a bee is released after three hours of waiting at a feeding spot, it takes off in the opposite direction from the nest, as if trying to get from the nest to the food source [62] but bees trained to two feeders at different times of day can still find their way back from a feeder even at times not associated with feeding [63]. This demonstrates that time-specific route memories related to time-specific rewards were still available and could be used at other times.

At a higher scale, one could ask about the relative importance of different strategies of foraging in dissimilar environments (Figure 1). In novel environments, path integration seems to overcome landmark information [64]. This suggests that landmark information may take longer to process or form in a salient manner, but path integration is a robust way of navigating over a short range. It has been suggested that all different strategies and features can be used simultaneously in a heterarchy, weighted by their salience and information content [65].

## 4. Orientation Flights and Returning to Known Locations

Bees perform a stereotypical flight associated with visual learning of a location. During nest orientation flights [66], and when learning the location of a food source upon departure [67,68], they face their goal and fly side-to-side and back-and-forth in a series of expanding arcs [69]. During these learning flights, they are thought to extract and memorize useful visual information about important cues. Subsequently, bees likely refer to these memories and compare the memorized scene with real-time visual input: to reach the goal, they can minimize the difference between the memory and the visual input [70]. The bee remembers not only the visual cues associated with the goal itself but the general composition of the immediate surrounding (learned during smaller and lower flight arcs) and even distant panoramic information (in the larger and higher flight arcs). The bee potentially associates a direction (compared to a reference point, such as *e*-vector) with the memories of that viewpoint. Orientation flights are important for bees that are new to an area [71]. Bees can find their way back to the hive from much farther than their expected area by using only a short learning flight near the hive [13]. When bumblebees leave one source and try to find another food source, they start by performing a learning flight at the first source and then an optimized expansion of the search area to find new flowers. This flight has a sequence of arcs with increasing diameters and these back-and-forth motions form loops [72,73]. The same behavior is observed during the first learning flights around the hive by new foragers and by misplaced bees trying to figure out where they are. When approaching a food source, bees learn the apparent size of nearby landmarks. However, it is only after the learning flight, during departure, that they also learn the distance to these landmarks [74]. Bees acquire the overall three-dimensional structure of the goal area during turn-back-and-look behavior [75] but they learn the color, shape, and size of landmarks during arrival at the food source.

When bees arrive at a target, they first aim at a landmark and associate it with their stored view. Their trajectory then brings them from the landmark to the goal so that the goal can be reached by image matching. They then move in a way such that the landmark matches the position and size on the retina that it had during the learning flights. [76]. Bees also use self-induced image motion to pinpoint the goal by matching the depth of field information computed from the parallax of different areas of their panorama [77]. Upon return, they approach the goal by matching the parallax from self-motion to the parallax during learning flights [41] in the absence of textural and contrast cues [78]. By controlling the image motion rate within a small range, bees ensure smooth flights with slower flight speed as they approach and land on their goal [79].

## 5. Different Sensory Modalities

### 5.1. Olfaction

Odors can inform honey bee searching for and returning to food sources [9]. For example, odors trigger recall of specific food sites by triggering odor-associated landmark memories [80]. Honeybees mark their nest entrance, as well as food sources, with Nasonov pheromone and other volatiles to attract other bees to these goals [81,82]. Similarly, stingless bees create and follow odor trails that consist of odors deposited every few meters [83]. In some cases, these odor trails may extend from the nest to the rewarding food source, but it seems that such trails provide an outbound route to food, not information directing bees back to their nest, though this remains to be tested [84]. Wasps migrating to a new nest site orient to odors deposited on the edges of leaves along the path to the new nest site [85]. Odor trails also play a major role in the orientation of multiple ant species and act as olfactory landmarks guiding workers to resources and back to the nest [86,87].

### 5.2. Magnetoreception and Mechanosensation of Electrical Charges

In the realm of landmarks that are located very close to or at the goal, there is evidence that bumble bees [88] and honey bees [89] orient via mechanosensory hairs to the electrostatic fields generated by floral resources and learn these fields, opening up the possibility that they could also use these senses to recognize their nests [90]. There is data suggesting that honey bees may sense magnetic fields [91,92,93,94] and that disrupting their ability to sense the earth’s magnetic field reduces their homing abilities [95]. Ants have been shown to actively use the magnetic field to orient towards their goals [96]. Additional studies would be revealing.

### 5.3. Multimodality

Landmarks and landscapes typically provide multiple types of information. Ants use both visual and chemical cues in their orientation towards food [97]. *Cataglyphis* ants can be trained to use visual, olfactory, magnetic, tactile, and vibrational landmarks to find their nest entrance [6,98]. Visual and olfactory cues play are important for allowing carpenter bees to recognize their nest aggregations [99]. Focusing on landmarks, stingless bees use both visual and chemical cues to orient towards rewarding food sources [100]. In bumble bees, the presence of a specific landmark odor can enhance discrimination of color cues and thereby the use of color as a local landmark [101]. Bumblebees learn the differences between floral scent patterns and subsequently transfer this learning to differences in visual patterns [102]. In these examples, the modalities likely have different saliences depending upon the species and context. For example, honey bees may pay more attention to the odors and colors of flowers than to other landmarks, although this varies with bee race [103]. Similarly, Gould and Towne [55] conclude that odors are more important than colors, and both are more important than landmarks and patterns, but others have suggested equal or independent effects of each type of stimulus [104]. 

## 6. Landmarks in Other Insects

The solitary bee (*Epicharis metatarsalis*) uses visual cues to find its nest entrance [105]. In stingless bees (*Tetragonisca angustula)*, homing bees orient towards the visual appearance of the nest from about 1 m away, likely corresponding to their ability to visually distinguish this landmark [106]. This example also illustrates the role of salience and reliability in information use.

Wasps perform orientation behaviors that are like the turn-back-and-look behavior of bees learning visual landmarks. Wasps also face the food item or nest site and fly back and forth in arcs that increase in radius over time. They will likewise circle high above the site to be learned while flying away, presumably to gain a better view of landmarks to use as snapshots [73,107]. 

Both wasps (Polistinae) and species of stingless bees (Meliponini) that deposit odor marks, tend to deposit these odor marks on conspicuous objects along the marked path [107,108,109]. These are examples of visually assisted chemotaxis in which insects modify a landmark with a signal to make it more conspicuous, just as multiple vertebrates species deposit scent marks on visually conspicuous landmarks [1]. Bees can also deposit cues, not just signals, on landmarks. Bumble bees and stingless bees leave footprint odor cues [83,110] on food sources, though, unlike signals, such marks are a byproduct of visitation.

Ants that follow odor trails shape their trails along edges and other features [86]. The way in which these physical edges shape the path is similar to how landscape edges can influence the flights of bees [9]. *Melophorus bagoti* ants can learn two different routes, one for the outbound and one for the inbound journey, and these routes are direction-specific [111]. Similarly, *Formica japonica* ants use landmarks differently if they are displaced at different points along their round trip [112]. These results suggest that the same landmarks or views trigger different responses depending on the context or motivational state. 

## 7. Neural Bases of Landmark Memory

How are features detected and memorized and used to guide future behavior? The concept of feature detectors as local neural circuits that process simple sensory inputs was suggested in the 1950s [113]. Relatively recently, the detailed anatomical and functional studies of the optic lobes of diverse insects (bees, ants, flies, etc.) have begun to find these feature detectors [114,115,116,117]. The circuitry of simple feature detectors in the eyes [118,119,120], memories of visual features in the mushroom bodies [121,122], and movement planning and representing spatial orientation in the central complex [123,124] provide promising research directions. 

In fruit flies, landmark orientation and path integration vector information are processed in the ellipsoid body, a part of the central complex [124]. In bumblebees, different populations of neurons in the lateral protocerebrum show preferential color and motion sensitivity [125], and some wide-field cells of the lobula can recognize the particular motion signatures of landmarks [120]. However, the mechanisms by which different headings are merged with different stimuli, including landmarks, when the animal is moving towards a goal, are not well understood in insects. Advances in neural recordings in virtual reality environments have opened up new possibilities for delineating the intricate interactions between different cues in simple and complex environments [126] and using such virtual reality can facilitate our study of learning in action in an extremely controlled manner [127,128,129] and allow us to test alternative hypotheses about representations of space, such as the cognitive graph [39], in bees. 

## 8. Recent Findings and Future Directions

As new neuroanatomical and behavioral data are discovered, new models have been built that closely mimic and account for the choices of navigating animals’ using realistic stimuli [20,130,131,132]. These models provide insight into the computations that navigating animals perform and allow us to test different hypotheses on the importance of visual features. In developing these models it is crucial to consider and balance the types of input bees see in the real world and how they actively respond to abstract and simplistic stimuli [133,134] that provide greater control but perhaps lesser relevance.

An important question is whether landmarks are remembered as part of a panorama, as separate entities, or as features. Bees learn to generalize features and categorize stimuli [135,136] and they may generalize the salient features of a landmark from different orientations and treat it as a unique entity.

How many different sets of snapshots and vectors can a bee learn? In an interesting example, receiving ambiguous polarization patterns drives bees to dance for multiple vectors even within the same dance [137] although perhaps navigational vectors are coded in a different way than waggle dance vectors [31,138]. Research on ants navigation should help elucidate this topic [139,140].

If we next consider the path that a bee takes, do they learn landmarks en route to a goal in the same way that they learn landmarks at or very close to the goal? Are floral memory and landmark memory separate processes? Do bees expect to see the next landmark in the sequence when they get close to where the vector from the previous landmark led them? These questions have been the topic of many studies, but definitive proof for some of the competing hypotheses is still lacking. Methodological advances in cognitive and physiological aspects of vision are paving the way for answering them. 

Finally, one might ask about the broader salience of understanding bee navigation. This information is useful, not only for understanding how bees navigate but, more generally, for understanding navigation in other animals, since mammals and insects share multiple mechanisms of feature detection [141]. Studying how bees navigate could help us understand the general mechanisms that have evolved to solve the problem of animal navigation and could bioinspire the development of better machine vision strategies for multiple applications, such as robots that navigate using view-based information and emulate the active vision strategies of animals [75]. Characterizing the continuum of landmarks and landscapes and the features used to identify locations will be a central part of this effort.

## Figures and Tables

**Figure 1 insects-10-00342-f001:**
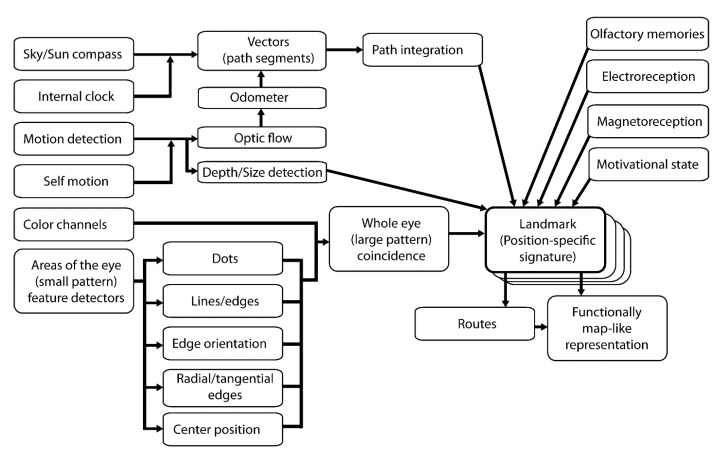
Different sensory stimuli and processing pathways for landmark detection and spatial navigation. At each location, the coincidence of multiple sources of sensory input is merged and combined into a signature. Each signature may be functionally attached to a navigational motor output such that a particular path is chosen by an animal in a specific motivational state. Multiple landmarks and routes may be combined to form a functional allocentric representation of the environment. Other cues and processing pathways play roles in guiding the navigation of bees, but we have limited our focus to the ones most studied.

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
