# Peer review of "The Role of Landscapes and Landmarks in Bee Navigation: A Review"

_insects, 2019, doi:10.3390/insects10100342_

Round 1

Reviewer 1 Report

I have multiple comments written into the odf of the ms, and I will send this to the authors directly. 

In addition I have the following more general comment:

Dear James and Bahram,

Many thanks for the draft of your review. I have made quite a few comments and I hope they will be helpful. Consider condensing your text, because I believe it includes multiple repetitions and lengthy parts that appear to me too speculative. I have not marked any of these parts and want to leave it for you, but please look particularly critical to the first third dealing with definitions. In my view there is too much semantics that does not help very much to design und carry out experiments. This appears to me important because at the end you are not coming up with a more useful definition of the term landmark.

I have marked a short paragraph in lines 423-427 that appears to me at the center of your arguments but have not been dealt with throughout the text. Rather you have followed the traditional way by assuming that data collected in close-up test conditions (within a few meters to a goal) are directly relevant for far ranging travels of the animal. For example, snapshot memories built in a close up situation are thought to be representative for panorama matching strategies in far ranging travels. But what is the evidence for that assumption? Don´t you think it is confusing and even misleading to talk about navigation under close-up test conditions? Is a landmark experienced during far ranging travels the same as feature in the cm range learned during learning flights directly at the goal? Do the exploratory orientations flights serve the same function as the learning flights at the goal? There are many more such questions which come up when questioning the tradition view advanced particularly by the Collett and Zeil group.

I would also have expected that you include in your discussion of the term landmark its object identity not only with respect to its visual (and other sensory) features but also its location irrespective how you define location (as the endpoint of a vector, as a character of the landscape during sequential experience, or in its spatial relation to other landmarks).

  As you will see am particularly unhappy with your reference to the Horridge book. This book is full of misinterpretations of data of other authors, of weak or uncontrolled data collected by Horridge himself which he was unable to get accepted for publication, and of conceptual weaknesses and failures. Just one example, he does not understand the difference between perception and sensation. I must say, I did not find anything helpful in this book on your topic either. I am afraid to say that you might do to yourself a questionable service by referring to this book.

You are using phrases like can learn, can use, can apply, can trigger….. frequently throughout the text. What does this “can” mean? Bees may not learn, use, apply, trigger…. under different conditions? Wouldn´t it be much better to simply say they learn, use, apply.

Reviewer 2 Report

Comments to the Authors

The terms ‘landscape’ and ‘landmark’ are commonly used in the navigation literature, and the authors first set out to define them, based on our current knowledge of how bees navigate. They follow it up with an up-to-date review of our knowledge on spacial encoding, navigational strategies, memory formation, and the neural bases of memory in bees.

I find the manuscript well-written and easy to follow. The authors tackle a difficult subject, but manage to explain the complexities in clear language. They find that definitions of the terms ‘landscape’ and ‘landmark’ are to some degree fluid and hard to pin down, as they depend on the point of view of the author and on other definitions. The introduction section does a good job of defining ‘landmarks’ in the broadest sense, including contexts (salience, motivation etc.) and sensory modalities (e.g., magnetoreception) that are often overlooked. It remains, however, a bit elusive where ‘landmarks’ end and ‘landscapes’ begin. This is indeed a continuum in natural settings, as the authors explain early on. However, I believe the manuscript, especially in the later sections on the different navigational strategies, could benefit from embracing this natural continuum, and trying to focus a bit less on a clear distinction between the two terms. In fact, many of the discussed strategies do not require this distinction, but are rather based on differential weighting of elements depending on their salience, utility or reliability. Landmark elements are then weighted strongly, and landscape elements are weighted less strongly, but still retain some influence. This is particularly important when discussing ‘snapshot’ memories or when talking about landmark recognition in different panoramic contexts (e.g., your L. 212-217). Many recent studies have tried to use such a more nuanced perspective by using terms such as “visual (guidance) cues”, “view-based memories”, or “panoramic views”.

In addition, I have some more detailed comments and suggestions that would need to be addressed:

L. 33: “… position of celestial objects above the dark edge of a forest in the skyline” The wording is a little unclear here. The dark edge of a forest is the skyline, right?

L. 152, 157, 315, 364: Is it really worth creating acronyms like SRM, GLM and TBL? Only two of them get a second mention in the text, and none a third.

L. 169-170: I couldn’t locate the section in the manuscript where you explain how other strategies could account for novel shortcuts, but I think it is very important to have. Can you add this?

L. 187 ff.: I understand that you want to differentiate between ‘alignment image matching’ and ‘positional image matching’ here, as they describe two different navigational behaviours (namely, keeping a heading and pin-pointing a position). However, as you explain nicely, the underlying strategy is essentially the same in both cases. The bees try to maximise the match between current and memorised views through corrective movements. In both cases, the current view needs to share some familiar features with the memory to work. You do differentiate between comparing ‘views’ for alignment matching, and comparing ‘relationships’ for positional matching, but how much evidence do we really have that ‘relationships’ are matched? Or that insects ‘extrapolate the current location’ (L. 191), for that matter? Rather than extrapolating the current location, one could readily deduce the direction to move in from retinotopic/snapshot images alone, which might be all it takes for positional image matching too.

L. 196: Consider replacing ‘calculate’ with ‘integrate’. This makes fewer assumptions as to how it is performed in the brain.

L. 205 ff.: Is it more likely that only features are extracted? There is quite a body of work (at least on navigating ants) that landscape features are part of their visual memories (not only in the sense of creating a context for extracted features). This links back to my main comment above.

L. 207-209: It might be worth citing the paper by Judd & Collett (Judd SPD & Collett TS. 1998. Nature 392:710-714) here, as it provided good evidence for the use of Cartwright & Collett’s snapshot model on routes (though on ants).

L. 273-286: I understand that you want to focus your manuscript on bees, but there is also a lot of work on feature hierarchies in ants (desert ants in particular). This should at least get a mention here. A starting point would be Hoinville & Wehner 2018. PNAS 115:2824-2829.

L. 317-320: This section seems a little out of place here. You mention this before anyway (in the sections on vectors and snapshots).

L. 321-325: Motion parallax is also used by bees to distinguish (‘extract’) a landmark from its background (Dittmar et al. 2010. JEB 213:2913-2923).

L. 366: “The will likewise circle high above the site to the learned while flying away…” Something went missing in this sentence, please correct.

L. 377-380: This section on M. bagoti is a bit jumbled or at least hard to understand. These ants do learn their (visually guided) way back to the nest. They are able to learn two different routes for the outbound and inbound journey, and these memories are not ‘reversible’. When the ant is in an ‘inbound state’, she will readily recognise the inbound route, but will not recognise its own outbound route in this state.

L. 382-383: I don’t think you can claim “These results suggest that views of certain landmarks may strictly trigger particular vectors” from the preceding results on M. bagoti and F. japonica. It’s quite the opposite, actually. Rather, it shows that the same landmarks (or views) can trigger different responses depending on the context or motivational state. (This may be vector-based, or this may be view-matching based.)

L. 395-397: It is not possible to understand from your description how the ellipsoid body is thought to store and compute heading directions and path integration. Please rephrase.

L. 418 ff.: The question of ‘how much can a bee remember’ is not really worth asking yet, as it is still pretty much unclear how visual memory is actually encoded in the brain (or even if it is in only located in the mushroom bodies, for that matter…). Memorising of visual information in distinct snapshots is only one model of visual memory in insects. There are some models that go beyond snapshots, e.g. Baddeley et al. 2011, Adaptive Behavior 19:3-15; or Wystrach et al. 2013, JEB 216:1766-1770. I would like to see this developed a bit more in this paragraph.

L. 425: I am not sure what you mean by “…so that the skyline is vertically at the centre of the visual field”. Can a skyline ever be vertical?

Reviewer 3 Report

Review of Kheradmand and Nieh manuscript

The paper provides a selected review of navigation in bees, especially honeybees. It distinguishes near and far landmarks, and provides a personal selection of literature and topics on bee navigation. Reviews on insect navigation in the most recent years have focused on ant navigation, so that this piece makes a welcome contribution. The authors provide a scholarly overview, and the writing is largely clear. I think that the piece is worth publishing, but have a few suggestions for improving it. My review will be biased with a distinct myrmecophile flavor from my own research experience, so that the authors should keep that in mind.

The first point concerns placing the review in context. One suggestion is to put in one or two reviews of ant navigation in the intro, such as Knaden and Graham’s 2016 Annu Rev Entomol paper. Cody Freas et al. (2019, Behav Proc) provide a review on learning to navigate, also focusing on ants, especially desert ants. As this learning theme also appears in the review, that’s another piece that could be worth citing. This latter piece also gives the authors a reason to now focus on bees learning to navigate in one of their sections. This kind of brief background would give a raison d’etre for focusing this piece on bees. With the authors’ focus on placing navigation using external objects into some kind of scheme, they might want to bounce off from, criticise, and develop from a piece that attempted to formulate a grand scheme. One piece that I can suggest stems from a Strungmann Forum, a chapter led by Jan Wiener including a host of authors who are experts in navigation in a range of animals, in the book called Animal Thinking, edited by Randolf Menzel and Julia Fischer (2011). It’s some years now since 2011, and developing the ideas there or reacting to ideas there is another way to place the current review in context. Without some kind of background context, the piece feels too much like it stands on its own, out of context.

When it comes to types of navigation, one recent theme from ant navigation might be worth mentioning and comparing with bees. The idea is that the animals oscillate left and right in travel, with the amount of oscillation controlled by various servomechanistic processes. These two basic units of action, oscillator and servomechanism, work hand in hand in navigation. When things look fine in travel, oscillations are small, and the animal mostly heads straight. When things look strange, unfamiliar, etc., then oscillations increase, and the animal meanders, or stops and looks around. Antoine Wystrach has developed this idea in various recent publications, really explicitly in Drosophila larvae (a rare publication from that author not on ants; 2016, eLife). A chapter by Kodzhabashev and Mangan models this process. I wonder how such ideas apply to bee navigation, especially honeybee navigation.

Other comments relate directly to contents of the review. I will list them one by one here with indications of approximate line numbers to which they apply.

141 landmark [singular]

160 the authors might want to compare and contrast a recent idea called cognitive graph, developed by Bill (William) Warren for humans. See, for example, his 2019 J Exp Biol publication. How does this idea work or not work for insects, honeybees in particular?

168 For such a simpler mechanism, it is worth checking out Thierry Hoinville and Wehner’s 2018 model, in PNAS. This is a sophisticated formal model that explains some bee data as well as ant data.

199 This idea of low error needs serious explaining. Errors are cumulative in path integration, so that I have no idea why you say "low error".

200 Ants learn a panorama with as little as one trip to a feeder. See the work of Cody Freas. High familiarity may not be needed or can be quickly acquired.

215 A whole lot more serves as context, including time of day. See work of Mario Pahl in the past decade. Srinivasan and co. also showed that motivational state (out foraging vs. homing) serves as contextual due, and did Fred Dyer and co. Cheng sees contexts as the basis for orchestrating various navigational servomechanisms. It's a huge theme.

232 On food vectors, and as you are relating ant work on this theme, the work of Harald Wolf deserves citation.

257 A parallel is found in vertebrates in the case of shape geometry vs. summary parameters such as principal and medial axes. See variously Brad Sturz, Debbie Kelly on this topic. But it might be too far off the track to include.

277 These are all part of what constitutes context.

310 Although the work is on wasps, scholarship on this theme demands that you cite and discuss the work of Jochen Zeil, classic work in the 90s and recent work.

343 Surely, you should mention the exciting recent discovery of the use of magnetic cues in the initial learning walks of ants. See the work of Pauline Fleischmann.

348 Don't forget tactile cues.

375 There is so much on ants that this bit does not do justice to the richness of that line of work. At the least, citing a bunch of recent references, esp. reviews, would be in order. Cody Freas et al.'s 2019 review provides an overview of all the different cues that ants use, plus it would cite other reviews.

394 ellipsoid body: I would specify that this is a structure in the central complex, which is turning out to be a navigational hub in the insect brain.

398 protocerebrum: again, part of the central complex

425 You mention skyline here for the first time. It deserves description earlier in the review, both in bees and ants.

431 See recent learning models applied to ant navigation, e.g., Wystrach et al's 2019 Anim Cogn paper.

Some references

Bolek, S., Wittlinger, M., & Wolf, H. (2012). Establishing food site vectors in desert ants. Journal of Experimental Biology, 215, 653-656.

Bolek, S., & Wolf, H. (2015). Food searches and guiding structures in North African desert ants, Cataglyphis. Journal of Comparative Physiology A, 201(6), 631-644.

Fleischmann, P. N., Christian, M., Müller, V. L., Rössler, W., & Wehner, R. (2016). Ontogeny of learning walks and the acquisition of landmark information in desert ants, Cataglyphis fortis. Journal of Experimental Biology, 219, 3137-3145.

Fleischmann, P. N., Grob, R., Müller, V. L., Wehner, R., & Rössler, W. (2018). The geomagnetic field is a compass cue in cataglyphis ant navigation. Current Biology, 28, 1-5.

Fleischmann, P. N., Grob, R., Wehner, R., & Rössler, W. (2017). Species-specific differences in the fine structure of learning walk elements in Cataglyphis ants. Journal of Experimental Biology, 220(13), 2426-2435.

Fleischmann, P. N., Rössler, W., & Wehner, R. (2018). Early foraging life: spatial and temporal aspects of landmark learning in the ant Cataglyphis noda. Journal of Comparative Physiology A, 204, 579–592.

Freas, C. A., & Cheng, K. (2018). Landmark learning, cue conflict, and outbound view sequence in navigating desert ants. Journal of Experimental Psychology: Animal Learning and Cognition, 44(4), 409-421.

Freas, C. A., Fleischmann, P. N., & Cheng, K. (2019). Experimental ethology of learning in desert ants: Becoming expert navigators. Behavioural Processes, 158, 181-191. doi: https://doi.org/10.1016/j.beproc.2018.12.001

Graham, P., & Wystrach, A. (2017). The emergence of spatial cognition in insects. In M. C. Olmstead (Ed.), Animal cognition: Principles, evolution, and development (pp. Chapter 4). Hauppauge NY: Nova Science.

Grob, R., Fleischmann, P. N., Grübel, K., Wehner, R., & Rössler, W. (2017). The role of celestial compass information in Cataglyphis ants during learning walks and for neuroplasticity in the central complex and mushroom bodies. Frontiers in Behavioral Neuroscience, 11, 226.

Hoinville, T., & Wehner, R. (2018). Optimal multiguidance integration in insect navigation. Proceedings of the National Academy of Sciences USA, 115(11), 2824-2829.

Kelly, D. M., Chiandetti, C., & Vallortigara, G. (2011). Re-orienting in space: do animals use global or local geometry strategies? Biology Letters, 7(3), 372-375. doi: 10.1098/rsbl.2010.1024

Knaden, M., & Graham, P. (2016). The sensory ecology of ant navigation: from natural environments to neural mechanisms. Annual Review of Entomology, 61, 63-76.

Kodzhabashev, A., & Mangan, M. (2015). Route following without scanning. Paper presented at the Conference on Biomimetic and Biohybrid Systems.

Pfeffer, S. E., Bolek, S., Wolf, H., & Wittlinger, M. (2015). Nest and food search behaviour in desert ants, Cataglyphis: a critical comparison. Animal cognition, 18(4), 885-894.

Sturz, B. R., Gurley, T., & Bodily, K. D. (2011). Orientation in trapezoid-shaped enclosures: Implications for theoretical accounts of geometry Learning. Journal of Experimental Psychology: Animal Behavior Processes, 37, 246-253.

Stürzl, W., Zeil, J., Boeddeker, N., & Hemmi, J. M. (2016). How wasps acquire and use views for homing. Current Biology, 26, 470–482.

Wehner, R., Hoinville, T., Cruse, H., & Cheng, K. (2016). Steering intermediate courses: desert ants combine information from various navigational routines. Journal of Comparative Physiology A, 202, 459–472.

Wiener, J., Shettleworth, S., Bingman, V. P., Cheng, K., Healy, S., Jacobs, L. F., . . . Newcombe, N. S. (2011). Animal navigation: A synthesis. In R. Menzel & J. Fischer (Eds.), Animal thinking: Contemporary issues in comparative cognition (pp. 51-76). Cambridge, MA, London: MIT Press.

Wolf, H., & Wehner, R. (2000). Pinpointing food sources: olfactory and anemotactic orientation in desert ants, Cataglyphis fortis. Journal of Experimental Biology, 203, 857-868.

Wolf, H., Wittlinger, M., & Bolek, S. (2012). Re-Visiting of Plentiful Food Sources and Food Search Strategies in Desert Ants. [Review]. Frontiers in Neuroscience, 6, article 102. doi: 10.3389/fnins.2012.00102

Wystrach, A., Lagogiannis, K., & Webb, B. (2016). Continuous lateral oscillations as a core mechanism for taxis in Drosophila larvae. eLife, 5, e15504.

Zeil, J. (1993a). Orientation flights of solitary wasps (Cerceris; Sphecidae; Hymenoptera) I. Description of flight. Journal of Comparative Physiology A, 172, 189-205.

Zeil, J. (1993b). Orientation flights of solitary wasps (Cerceris; Sphecidae; Hymenoptera) II. Similarities between orientation and return flights and the use of motion parallax. Journal of Comparative Physiology A, 172, 207-222.

Round 2

Reviewer 1 Report

The ms has been adequately dealt with the feedback from the reviewers. Many thanks for this work. It is still a bit too long, and this makes it less attractive reading. As you might imagine I do not agree with some of your interpretations concerning the relevance of experiments in the close space for navigation to those in the large space, but I accept your view as a summary of the traditional view dominated by a group of influential researchers. 

l. 414: please think about changing this sentence. In my view the cited paper gives no evidence to landmarks being stored or coded in the central complex. It only shows (in very reduced form) visual stimuli being connected to vector based neural activities in central complex neurons. 

l. 416: the lateral protocerebrum does not belong to the central complex

I like to ask you to include two references to balance the views you presented: l. 286 please make reference to the review of Vargues-Weber, Deisig, Giurfa (2011) Ann Rev. Entomol. 56, 423-443.

l. 363 please make reference to Martin and Lindauer (1977) J. comp. Physiol 122, 145-187  and to Walker and Bitterrman 1985 J. comp Physiol. A, 157, 67-71

I will deal with the topic discussed in lines 182ff and 448 in my review for this special issue in INSECTS. If you like you may refer to it.

Author Response

Thank you for the suggested edits. We have now incorporated the suggested references and modified the wording of the mentioned sentences, marked in cyan highlights. The yellow highlights are the previous changes to the major revisions.

l. 414: please think about changing this sentence. In my view the cited paper gives no evidence to landmarks being stored or coded in the central complex. It only shows (in very reduced form) visual stimuli being connected to vector based neural activities in central complex neurons.

- Thank you for this suggestion. We changed the word "stored" to "processed" as we agree that there is no evidence of information being stored in the ellipsoid body.

l. 416: the lateral protocerebrum does not belong to the central complex

We removed the phrase "also a part of the central complex".

I like to ask you to include two references to balance the views you presented: l. 286 please make reference to the review of Vargues-Weber, Deisig, Giurfa (2011) Ann Rev. Entomol. 56, 423-443.

-Reference added with a small comment.

l. 363 please make reference to Martin and Lindauer (1977) J. comp. Physiol 122, 145-187 and to Walker and Bitterrman 1985 J. comp Physiol. A, 157, 67-71

-Suggested references added.

I will deal with the topic discussed in lines 182ff and 448 in my review for this special issue in INSECTS. If you like you may refer to it.

- We would like to ask the editors to add a phrase referring to your review by mentioning your review title in the appropriate places, marked by the red highlight at the end of section 2, and at the end of the penultimate paragraph of section 8 : For a detailed discussion on this topic, refer to the review "__review title___" by "__author__" in this issue. Please let us know if there is a problem with this request. Since we do not know officially the identity of the reviewer nor the appropriate citation, we cannot insert this information.